# The syntenic long non-coding *RNA DANCR* is an essential regulator of zebrafish development and a human melanoma oncogene

Stephanie M. E. Jones[1], Elizabeth A. Coe[1¤], Michael Shapiro[1], Kelli M. Gallacher[1], Karen Camargo Sosa[1], Nikolas Nikolaou[2], Igor Ulitsky[3], Robert N. Kelsh[1], Keith W. Vance[1]*

1 Department of Life Sciences, University of Bath, Bath, United Kingdom, 2 Living Systems Institute and Department of Clinical and Biomedical Sciences, University of Exeter, Exeter, United Kingdom, 3 Department of Immunology and Regenerative Biology and Department of Molecular Neuroscience, Weizmann Institute of Science, Rehovot, Israel

¤ Present address: Oxford PharmaGenesis, Cardiff, UK

* k.w.vance@bath.ac.uk

## Abstract

Long non-coding RNAs (lncRNAs) play crucial roles in regulating gene expression. Some are essential for organismal development and physiology, and they can contribute to diseases including cancer. Whilst most lncRNAs exhibit little sequence similarity, conservation of lncRNA transcription relative to neighbouring protein-coding genes suggests potential functional significance. Most positionally equivalent lncRNAs are uncharacterized and it remains unclear whether they exert similar roles in distant species. Here, we identified melanoma-associated lncRNAs predicted to be components of the MITF gene regulatory network in human melanoma that have positionally equivalent transcripts in zebrafish. We prioritized the cancer-associated lncRNA *Differentiation Antagonizing Non-Protein Coding RNA* (*DANCR*) as an exemplar for functional investigation. *DANCR* is a multi-exonic, cytoplasmically-enriched lncRNA and small RNA host gene transcribed from syntenic regions in the human and zebrafish genomes. MITF and c-MYC, key melanoma transcription factors, regulate human *DANCR* expression and melanoma patients with high *DANCR* display significantly decreased survival. *DANCR* is a melanoma oncogene that controls cancer-associated gene expression networks to promote human melanoma cell proliferation and migration. Zebrafish *dancr* is essential for embryonic development. It is dynamically expressed across multiple different cell types in the developing embryo, transcriptionally activated by *mitfa* during early zebrafish development and it regulates genes involved in cell death. Our work suggests that cancer-critical lncRNAs such as *DANCR*, expressed from similar regions in vertebrate genomes, may control related genes and processes involved in both embryonic development and tumorigenesis across species.

**Data availability statement:** The datasets produced in this study are available in the following databases: Human siDANCR RNA-Seq data: Gene Expression Omnibus GSE292491 (https://www.ncbi.nlm.nih.gov/geo/query/acc.cgi?acc=GSE292491) Zebrafish dancr crispant RNA-Seq data: Gene Expression Omnibus GSE292918 (https://www.ncbi.nlm.nih.gov/geo/query/acc.cgi?acc=GSE292918). Numerical data underlying the Figs are shown in S6 Table.

**Funding:** This project has been funded by a Biotechnology and Biological Sciences Research Council (BBSRC) Southwest Biosciences Doctoral Training Partnership PhD studentship (KWV, RNK, SMEJ) and BBSRC grants awarded to KWV (BB/N005856/1; KWV, MS) and NN (BB/Y009533/1; SMEJ, NN). The funders had no role in study design, data collection and analysis, decision to publish, or preparation of the manuscript.

**Competing interests:** The authors have declared that no competing interests exist.

## Author summary

LncRNAs are a class of gene expression regulators, with suggested roles in development and disease, including cancer. Despite limited DNA sequence similarity, the conserved positioning of lncRNA genes in different genomes has been proposed to indicate functional importance across species. However, many of these lncRNAs have yet to be studied and their biological relevance remains unknown. Here, we identified candidate melanoma-associated lncRNAs that act within a key cancer network in humans and that are expressed from equivalent locations in the zebrafish genome. We focus on one of these, called *DANCR*, and show that it is a cytoplasmically enriched lncRNA that also contains two functionally important small RNAs within its sequence in both human and zebrafish. We reveal that human *DANCR* is regulated by the cancer causing MITF and c-MYC transcription factors in melanoma, promotes human melanoma cell proliferation and migration and is associated with poor patient survival. Zebrafish *dancr* is essential for embryonic development and regulates cell death genes. Our findings suggest that positionally equivalent lncRNAs such as *DANCR* may perform related roles in development and cancer across different species.

## Introduction

Vertebrate genomes are extensively transcribed, expressing tens of thousands of long non-coding RNAs (lncRNAs) that do not encode proteins. LncRNAs are a functionally diverse class of molecules that regulate multiple molecular processes including gene transcription, chromatin modification, RNA processing, splicing, editing, localization and stability, as well as protein synthesis and localisation [1]. Some lncRNAs exert important biological functions in development and normal physiology and have visible phenotypes when mutated in mice models [2–4]. However, the importance of most lncRNAs remains unknown and many continue to be considered non-functional transcriptional noise [5].

LncRNA sequence and transcription are rapidly turned over during evolution with only approximately 5% of lncRNA sequence, mostly lncRNA promoter regions and exonic splice enhancer (ESE) motifs, remaining conserved between vertebrates [5–8]. Despite this, large numbers of lncRNAs display dynamically regulated expression during development and are predicted to be biologically important. Moreover, the expression of syntenic lncRNAs from equivalent regions in distant genomes relative to their neighbouring protein coding genes can indicate functional significance. For example, the lncRNA *Paupar* is transcribed upstream from the *Pax6* transcription factor gene in human, mouse, dog, frog and zebrafish. Mouse *Paupar* is co-expressed with *Pax6* in the neural lineage and is required for postnatal neurogenesis *in vivo* and neuroblastoma cell proliferation and differentiation *in vitro* [9–11]. The human *PAUPAR* orthologue is essential for cortical differentiation of embryonic stem cells and both human and mouse *Paupar* interact with PAX6 to regulate neural

gene expression suggesting conserved mechanisms of action [11–13]. Despite the rapid evolutionary turnover of their sequence, which may be consistent with only short sequences being required for function, the number of lncRNAs with essential functions in embryonic development and organismal physiology may therefore be greater than that predicted based on primary sequence similarity alone.

Comparative analysis of genomic and transcriptomic datasets has discovered syntenic orthologues of mammalian lncRNAs in distantly related vertebrate genomes. 570 human lncRNAs were predicted to have zebrafish orthologues based on conserved genomic locations and patterns of RNA-binding protein interactions [14]. Depletion of four of these resulted in developmental delays in zebrafish embryos and a reduction in proliferation in human cancer cells. Microsynteny analysis also identified 16 positionally conserved intergenic lncRNAs between amphioxus and human genomes [15]. One of these, *Hotairm1*, is similarly expressed in the anterior portion of the neural tube in both amphioxus and frog and is required for development of the anterior part of the central nervous system in frogs. Whilst orthologous lncRNAs with similar functions can thus be identified based on syntenic transcription, positional equivalence does not necessarily imply biological function. Deletion of 25 syntenic lncRNAs in zebrafish, many of which were located in close genomic proximity to developmental regulatory genes, revealed that none of these lncRNAs were necessary for maintaining outwardly normal embryogenesis, viability or fertility [16]. Individual lncRNAs therefore need to be prioritised and investigated on a case-by-case basis to define their relative importance in development and disease.

Large-scale loss-of-function screens have suggested that some lncRNAs are essential for the growth of transformed cell lines in culture. CRISPR-Cas9 editing of lncRNA splice sites identified 230 lncRNAs as essential for the growth of chronic myeloid leukaemia K562 cells whilst CRISPRi-mediated repression of lncRNA transcription demonstrated that 499 human lncRNA loci are required for the growth of at least one transformed or stem cell line [17,18]. Indeed, mutations and alterations in lncRNA genes have been shown to contribute to the genetic susceptibility to cancer. LncRNA genes are frequently amplified or deleted in cancer and some function as part of oncogene and tumour suppressor gene regulatory networks to control cancer hallmarks in multiple different tumour types [19,20].

The microphthalmia-associated transcription factor (MITF) plays a critical role in melanocyte development and in melanoma [21]. MITF regulates genes important for proliferation, differentiation, senescence, invasion, metastasis and metabolism [21] and several lncRNAs act within the MITF network in melanoma. *DIRC3* is a MITF-repressed lncRNA that blocks the anchorage-independent growth of melanoma cells whilst the MITF activated lncRNA *LENOX* promotes melanoma cell survival and resistance to MAP kinase inhibitors [22,23]. On the other hand, *TINCR* acts upstream of MITF to repress its expression and block the spread of melanoma whilst *SAMMSON* is frequently co-amplified with MITF in melanoma and is essential for melanoma cell proliferation and survival [24,25]. LncRNA components of the MITF network are thus predicted to be important regulators of melanocyte development and melanoma biology.

In this study, we used positional synteny to annotate lncRNA components of the MITF network in human melanoma which may also have conserved functions in vertebrate development. We discovered that transcription of the cancer-associated lncRNA *Differentiation Antagonizing Non-Protein Coding RNA (DANCR)* relative to its neighbouring protein-coding genes is robustly conserved throughout vertebrate evolution and prioritised it as an exemplar for functional investigation. We show that both human and zebrafish *DANCR* are multi-exonic, cytoplasmically-enriched lncRNAs that host biologically important small RNAs within separate introns and control related genes and pathways involved in cell cycle and cell death. The results demonstrate that human *DANCR* is a MITF and c-MYC regulated lncRNA oncogene that promotes melanoma cell proliferation and migration and that melanoma patients with high *DANCR* expression have significantly decreased survival rates. We found that zebrafish *dancr* is essential for early embryonic development. It is dynamically expressed across multiple tissues and cell types in the developing embryo and activated by *mitfa* during early zebrafish development. Cancer-associated lncRNAs expressed from equivalent regions in vertebrate genomes, exemplified by *DANCR*, may therefore act as conserved regulators of both embryonic development and tumorigenesis.

## Results

### *DANCR* is a candidate melanoma-associated lncRNA expressed from positionally equivalent genomic locations in vertebrates

Here, we identified lncRNAs expressed from equivalent regions in the human and zebrafish genomes in which the human orthologue is expressed in melanoma and targeted by the key MITF transcription factor. These represent candidate melanoma associated lncRNAs that may have important functions both in the development of the neural crest and other lineages and in melanoma biology. We assembled a set of 11,881 melanocyte and melanoma expressed human and 11,511 zebrafish lncRNA transcript models using publicly available RNA-seq data and annotated 2,796 syntenic human lncRNA that have a positionally equivalent lncRNA transcript in zebrafish (S1 Table). 506 of these had promoters or genomic loci bound by MITF in human melanoma cells (Fig 1A; S2 Table), including the *LINC00520* and *LINC00673 (SLNCR1/slincR)* orthologous lncRNAs that have previously been implicated in human melanoma and zebrafish biology [26–29].

One of these lncRNAs, *DANCR*, was prioritised for functional investigation for the following reasons: (1) The genomic neighbourhood encompassing *DANCR* shows strong synteny amongst diverse vertebrates (Fig 1B). Positionally equivalent multi-exonic *DANCR* transcripts are expressed from orthologous regions in multiple vertebrate genomes, including human, mouse, chicken, zebrafish and elephant shark (Fig 1B), and in a similar direction relative to the neighbouring protein-coding genes. (2) Human *DANCR* is required for somatic progenitor cell self-renewal and acts as an oncogene in several different cancers [30–33]. However, its importance in melanoma and role in embryonic development is not well defined. (3) *DANCR* is a small RNA host gene in vertebrates. The locus contains two small nucleolar RNAs (snoRNAs) in zebrafish and a miRNA and snoRNA in humans within separate introns (Fig 1C). As 30 out of all syntenic lncRNAs that we identified overlapped a snoRNA, *DANCR* may be representative of a wider subclass of lncRNAs that host small RNAs within their introns. (4) The *DANCR* transcript is cytoplasmically enriched in both human melanoma cells (Fig 1D) and zebrafish embryonic cells (Fig 1E), suggesting that human and zebrafish *DANCR* may exert similar functions or work using similar mechanisms of action.

### *DANCR* is a clinically important MITF and c-MYC regulated lncRNA in human melanoma

Human *DANCR* is a growth promoting oncogene in several different cancers [34,35]. However, its regulation and functional importance in melanoma is poorly defined. Publicly available ChIP-seq data show that the *DANCR* promoter is bound by MITF [36] and c-MYC (ENCODE) (Fig 2A). These are the two most highly expressed bHLH-Zip transcription factors in the melanocyte lineage and are known regulators of genes involved in development and cancer [37]. To test if *DANCR* is directly regulated by MITF and c-MYC in melanoma we depleted these two transcription factors in multiple human melanoma cell lines using siRNA transfection and measured changes in gene expression using RT-qPCR. MITF knockdown led to a significant increase in *DANCR* levels in 501mel and decrease in A375 cells (Fig 2B). siRNA transfection did not significantly reduce MITF in SK-MEL-28 cells and no significant changes in *DANCR* were observed (Fig 2B). c-MYC depletion significantly decreased *DANCR* expression in 501mel, A375 and SK-MEL-28 cells (Fig 2C). MITF thus regulates *DANCR* in a melanoma cell dependent manner whilst c-MYC activates *DANCR* across melanoma cells. *DANCR* may therefore act within these cancer critical gene regulatory networks to control melanoma growth and metastasis. According, *DANCR* is highly expressed across all four melanoma genomic subtypes (NRAS, NF1, BRAF, and triple negative) in The Cancer Genome Atlas (TCGA) genomic and transcriptomic data (Fig 2D) [38] and in all melanoma cell subpopulations within a heterogeneous tumour model [23]. Moreover, analysis of TCGA survival data using OncoLnc [39] shows that melanoma patients with high *DANCR* expression have significantly decreased survival (Fig 2E). *DANCR* may play an essential role in melanoma downstream of MITF and c-MYC irrespective of tumour genomic status.

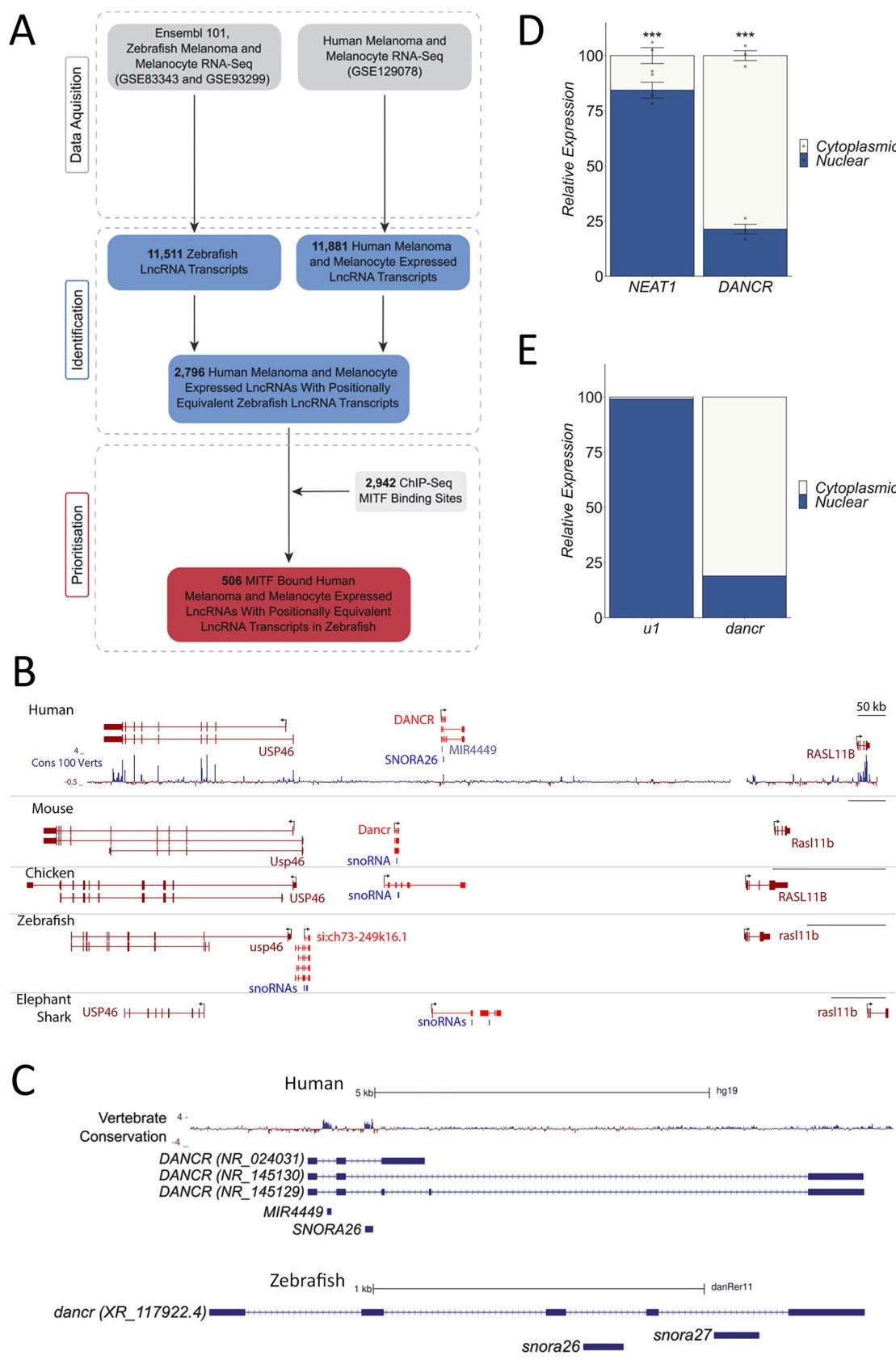

**Fig 1. *DANCR* is a candidate melanoma associated lncRNA expressed from equivalent genomic locations in vertebrates. (A)** Workflow used to identify human MITF-bound melanocyte and/or melanoma expressed lncRNAs that are transcribed from equivalent regions in the zebrafish genome. **(B)** *DANCR* lncRNA is expressed from syntenic regions in vertebrate genomes in an equivalent direction relative to the neighbouring *USP46* and *RASL11B* protein-coding genes. **(C)** *DANCR* is a multi-exonic lncRNA that contains two snoRNAs embedded within separate introns in the human (GRCh37/hg19) and zebrafish (GRCz10/danRer11) genomes. **(D, E)** *DANCR* is cytoplasmically enriched in both human and zebrafish. Human SK-MEL-28 melanoma cells **(D)** and zebrafish embryonic cells **(E)** were biochemically separated into cytoplasmic and nuclear fractions. The relative levels of *DANCR* and the indicated nuclear transcript controls were determined in each fraction by RT-qPCR.

### *DANCR* promotes human melanoma cell proliferation and migration

To investigate the role of *DANCR* in melanoma we first defined the transcriptional response to *DANCR* loss-of-function. Transient transfection of a *DANCR* targeting siRNA reduced *DANCR* transcript levels by ~85% in SK-MEL-28 cells. This resulted in significant changes in the expression of 98 genes compared to a non-targeting control (DESeq2 padj<=0.05; log2FoldChange>=0) (Figs 3A and S1A). Forty-three of these genes, including *DANCR* itself, were significantly downregulated whilst 55 genes were upregulated (S3 Table). RT-qPCR profiling of four key *DANCR* target genes in A375 melanoma cells following *DANCR* depletion using two siRNAs led to changes in expression consistent with the SK-MEL-28 RNA-seq data (S1B Fig), validating the specificity of these results. Gene Ontology (GO) analysis revealed that *DANCR* target genes in SK-MEL-28 cells are enriched for regulators of cell migration, cell adhesion, phosphorylation, response to stress, cell proliferation, programmed cell death and cell cycle and include genes such as *CDKN1A*, *CCND1*, *GAS1*, *CASP7* and the *ALDH1A3* melanoma stem cell marker (Fig 3B). Kyoto Encyclopaedia of Genes and Genomes (KEGG) pathway annotation showed that *DANCR* regulated genes are involved in cancer associated signalling pathways such as phosphoinositide 3-kinase (PI3K)-AKT, focal adhesion and p53 signalling (Fig 3C). *DANCR*-regulated processes and pathways are involved in both cancer and normal development and the results suggest that human *DANCR* may act to control melanoma cell proliferation and migration.

To test this, we determined the effect of silencing *DANCR* on cell behaviour using growth and wound healing assays. Depletion of *DANCR* by approximately 50–75% using transient transfection of two independent siRNAs reduced both the growth and migratory capacity of SK-MEL-28 cells compared to a non-targeting control (Fig 3D, E, F, G, H). *DANCR* depletion using siRNA did not affect expression of the embedded *SNORA26* gene in melanoma cells (S1C Fig), suggesting that the mature *DANCR* transcript is responsible for regulating these phenotypic changes. *DANCR* regulation of proliferation and migration was then corroborated in A375 cells using CRISPR interference (CRISPRi) (S2 Fig). Recruitment of the catalytically inactive dCas9-KRAB transcriptional repressor to the *DANCR* promoter using two different single guide RNAs (sgRNAs) silenced *DANCR* transcription by approximately 70–80% compared to a non-targeting control and also led to a significant reduction in A375 cell proliferation and migration. Altogether, the evidence suggests that human *DANCR* is a positionally conserved, clinically relevant oncogene that acts in a transcript-dependent manner to regulate gene expression programmes promoting proliferation and cell migration in melanoma.

### *Dancr* is dynamically expressed across multiple tissues and cell types in the developing zebrafish embryo

We next investigated the function of the positionally equivalent *dancr* locus in zebrafish development. To do this, the temporal expression profiles of *dancr* and its two embedded snoRNAs, *snora26* and *snora27*, were determined in whole wild type AB zebrafish embryos. RT-qPCR revealed that all three are expressed in all time points studied from 10-30 hours post-fertilisation (hpf) and that variations in the expression of *dancr* do not correlate with changes in either snoRNA (Fig 4A). RNAscope *in situ* hybridisation (ISH) was then performed in *Tg(Sox10:Cre)ba74; Tg(hsp70l:loxP-dsRed-loxP –Lyn-Egfp)* transgenic zebrafish to define the spatiotemporal expression of *dancr* during embryogenesis and enable lineage tracing of neural crest cells (NCCs) and their derivatives [40,41]. The results showed that *dancr* is widely expressed in all developmental stages studied (18, 22, 30 hpf) (Fig 4B, C, D). At 18 hpf *dancr* is expressed in the trunk and at lower

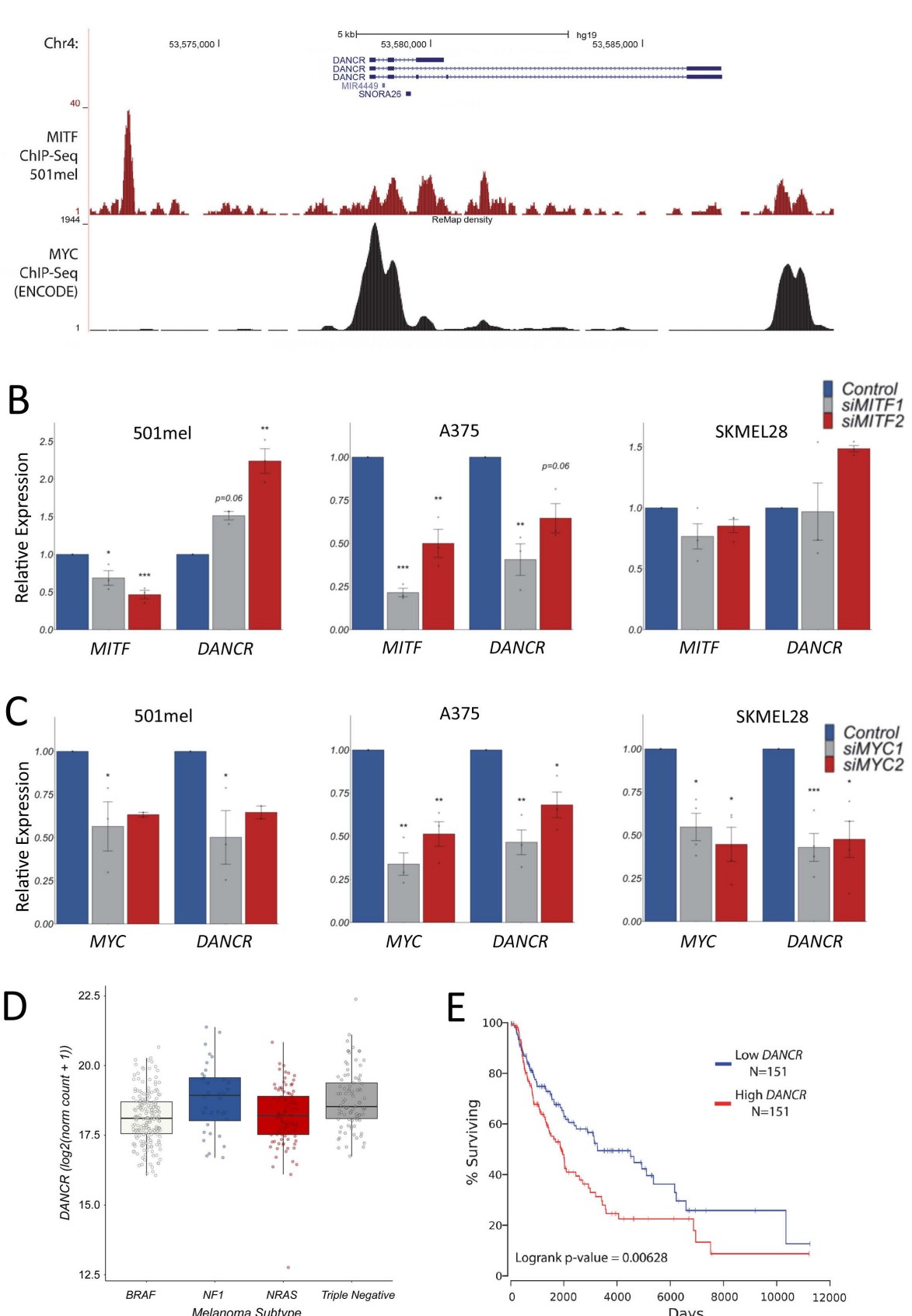

**Fig 2. *DANCR* is a clinically important MITF and c-MYC regulated lncRNA in human melanoma. (A)** UCSC genome browser view showing MITF and c-MYC ChIP-seq peaks across the *DANCR* locus (GRCh37/hg19). **(B)** MITF and **(C)** c-MYC regulate *DANCR* in melanoma. siRNA transfection was used to deplete MITF and c-MYC levels in 501mel, A375 and SK-MEL-28 cells. Expression changes were analysed using RT-qPCR. *POLII* was used as a reference gene. Results presented as mean +/- SEM., n ≥ 2; Two-tailed two sample t-test p < 0.05*, p < 0.01**, p < 0.001***. Individual dots represent separate biological replicates. **(D)** *DANCR* is highly expressed in all genomic subtypes of melanoma. Box and whisker plot (min to max; showing all points) representing *DANCR* expression in *NRAS*, *NF1*, *BRAF*, and triple negative genomic subtypes of melanoma using TCGA Skin Cutaneous Melanoma (SKCM) RNA-seq datasets. **(E)** Melanoma patients with high *DANCR* expression have significantly decreased survival. TCGA SKCM patients were sorted based on *DANCR* levels. Percent survival was compared between *DANCR* high (top third) and *DANCR* low (bottom third) groups. Cox regression analysis shows that high *DANCR* expression correlates with statistically significant decreased survival (logrank p-value = 0.00628).

levels in the GFP positive cranial NCCs (Fig 4B). At 22 hpf, *dancr* is expressed across the embryo and is most prominent in the posterior trunk, anterior tail, forebrain, eye and in the cranial and trunk NCC populations (Fig 4C). By 30 hpf, *dancr* expression becomes more restricted and *dancr* transcripts are predominantly found in the posterior trunk, anterior tail, and in the eye, including both the lens and retina (Fig 4D). There is a visible reduction in *dancr* levels in the head and forebrain at 30 hpf compared to 22 hpf and in the number of *dancr* expressing cranial and vagal NCCs (Fig 4C, D). *Dancr* is also expressed in the somitic blocks, but not in the notochord (Fig 4D), and co-expressed with *mitfa* in a subset of *sox10* positive presumptive melanoblasts in the tail region of the zebrafish embryo at 30 hpf (Fig 4E). This suggests that human and zebrafish *DANCR* are conserved components of the MITF network in melanocyte development and melanoma. Consistently, a consensus M-box motif predicted to be bound by MITF is located <1kb downstream zebrafish *dancr* locus (S3A Fig) and *dancr* expression is significantly reduced in homozygous mutant *nacre* whole zebrafish embryos that have a loss-of-function mutation in *mitfa* and lack neural crest derived melanocytes (Fig 4F). Furthermore, RNAscope also showed that *dancr* transcripts are mainly located in the cytoplasm in zebrafish embryos at 30 hpf and that nuclear *dancr* foci can also be detected in a small number of cells (Fig 4G). This agrees with the fractionation experiments in human and zebrafish cells and suggests that *dancr* may act both post-transcriptionally and as a direct regulator of gene transcription.

## *Dancr* regulates genetic pathways involved in cell death

Despite limited sequence similarity (S3A Fig), the dynamically regulated expression of a positionally equivalent *dancr* lncRNA in the zebrafish embryo suggests that the locus may be important for development as well as cancer. To study this, the *dancr* promoter was deleted using CRISPR-Cas9 genome editing with a pool of four guide RNAs (Fig 5A, S3B, C Fig), and F0 crispant zebrafish embryos were analysed at 24 or 30 hpf. RNA-seq showed that deletion of the *dancr* promoter effectively reduced transcription across most of the *dancr* locus, with the exception of the last exon, in *dancr* del F0 crispants compared to mock control zebrafish at 30 hpf (Fig 5A). This led to a significant decrease in the levels of *dancr*, *snora26* and *snora27* as evaluated by RT-qPCR (Fig 5B). Furthermore, *dancr* promoter deletion resulted in significant changes in the expression of 164 genes (DESeq2 padj<=0.05; log2FoldChange>=1.0) (Figs 5C, S3D and S4 Table). 124 genes are up-regulated and 40 are down-regulated in F0 crispants compared to control zebrafish. The *dancr* regulated gene set is significantly enriched in KEGG pathways involved in necroptosis, C-type lectin receptor signalling, Herpes simplex virus 1 infection, p53 signalling, steroid biosynthesis and Toll-like receptor signalling pathway (Fig 5D). These pathways regulate programmed cell death and the immune response and include key developmental and cancer-associated genes such as *cdkn1a, casp8, mdm2, bcl3, pou2f2a, irf9, gad45aa* and *mmp9* as well as the *cbx7a* component of the PRC1 complex which showed the greatest change in expression upon *dancr* loss-of-function.

## *Dancr* is essential for zebrafish embryogenesis

Deletion of the *dancr* promoter led to a large increase in the number of abnormal embryos (n = 163/292) compared to uninjected wild type sibling (n = 28/677) and mock (n = 23/220) control embryos at 24 hpf (Fig 6A, B). Multiple overt developmental defects were observed specifically in *dancr* del F0 crispants (Fig 6A) and phenotypic abnormalities were separated

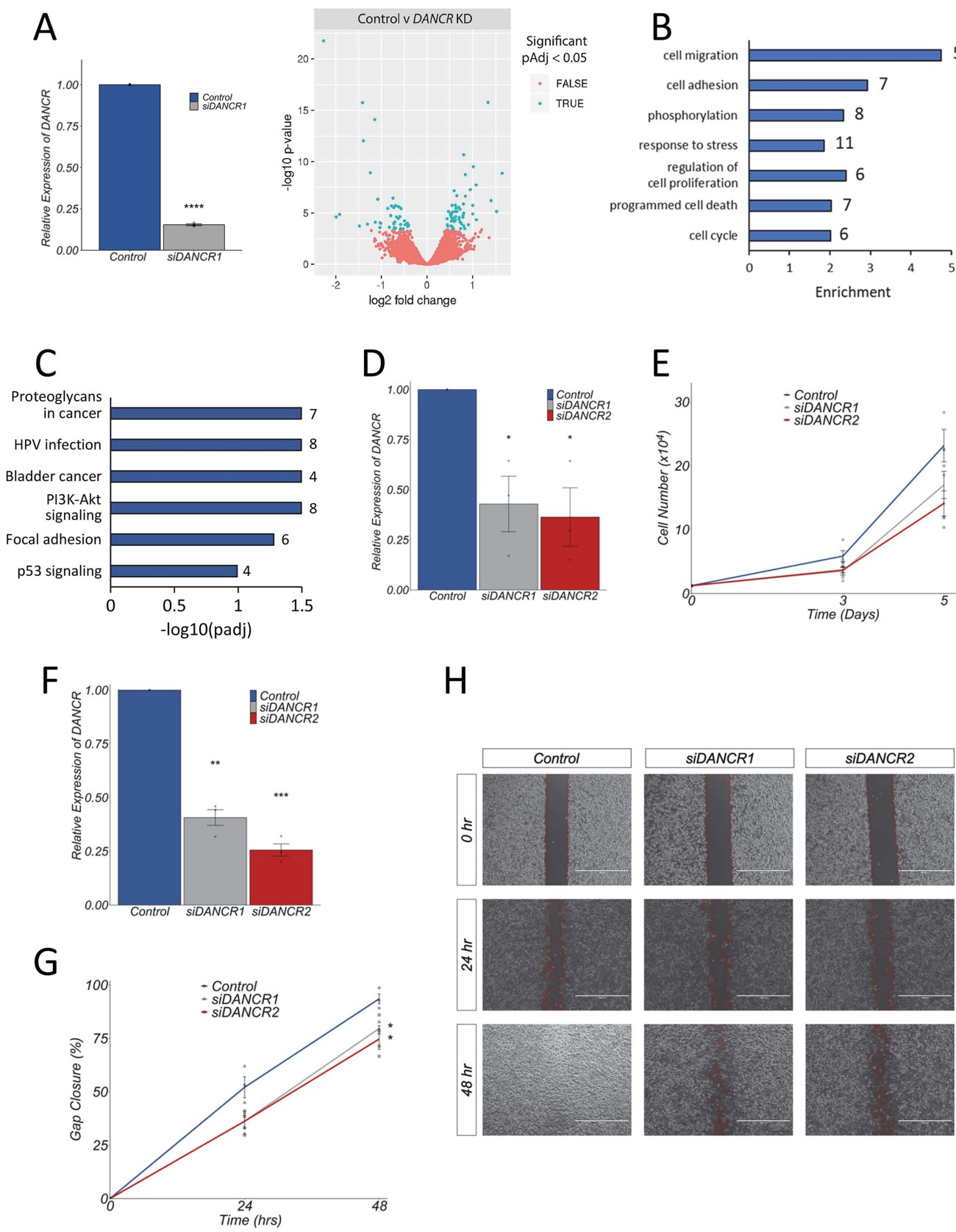

**Fig 3. *DANCR* promotes human melanoma cell proliferation and migration. (A, B, C)** *DANCR* target genes are enriched for regulators of cancer associated processes and pathways including cell proliferation, migration and death. **(A)** siRNA mediated *DANCR* depletion in SK-MEL-28 cells induces statistically significant changes in the expression of 98 genes compared to a non-targeting control (DESeq2 padj<=0.05; log2FoldChange>=0). GO enrichment **(B)** and KEGG pathway **(C)** analysis of the *DANCR* regulated gene set was performed. Representative significantly enriched categories (FDR 5%) and pathways (FDR 10%) are shown. The number of genes found in each category are indicated. **(D, E, F, G, H)** *DANCR* promotes melanoma cell proliferation and migration. *DANCR* levels were depleted in SK-MEL-28 cells using two independent siRNAs. Three days later *DANCR* expression was determined using RT-qPCR **(D, F)** and proliferation **(E)** or wound healing **(G, H)** assays set up. For proliferation analysis, cells were seeded in a 6-well plate and the total number of cells were counted at days 0, 3 and 5 **(E)**. For wound healing assays, cells were first treated with mitomycin-C to block cell proliferation and migration was determined using Ibidi chambers. The gap was imaged at 0, 24 and 48 hours and percent gap closure calculated using the ImageJ Wound Healing plugin **(G, H)**. Statistical analysis was performed at the 48-hour time point. For all RT-qPCR experiments, expression changes are shown relative to a non-targeting negative control siRNA (set at 1). *POLII* was used as a reference gene. All results presented as mean +/- SEM., n = 3. Two-tailed two sample t-test p < 0.05*, p < 0.01**, p < 0.001***. Individual dots represent separate biological replicates.

into four classes based on their severity (Fig 6C). The F0 *dancr* del crispant embryos had defects in the anterior to posterior axis, with the most severe Class I embryos lacking defined trunk and tail tissues and showing a reduction in head size, although anterior brain and eyes are present. Class II F0 *dancr* crispant embryos had identifiable trunk tissue, but with the tail poorly defined, and were generally underdeveloped and lacked somite patterning. Necrosis was prominent in both Class I and Class II embryos. Class III F0 *dancr* crispant embryos had a partial disruption to the tail which was either bent or coiled and a reduced head size; somites were present, but were block-like, lacking the classic V-shape. Class IV *dancr* del crispant embryos appeared to be phenotypically normal at 24 hpf and showed no discernible abnormalities. 70–74% of *dancr* del crispant embryos have a visible Class I-III phenotype at 24 hpf with approximately 12–30% of these being categorised as Class I with the most severe abnormalities (Fig 6C, D). Co-injection of 800 pg *in vitro* transcribed mature *dancr* lncRNA that has the snoRNA containing introns removed in *dancr* del crispants significantly (two-tailed two sample t-test; p = 0.01084) reduced the proportion of embryos with the most severe phenotype (Fig 6C). This partial rescue suggests that the observed loss-of-function phenotype is mediated at least in part by the fully processed *dancr* lncRNA transcript. Conversely, co-injection of 300 and 800 pg *snora26*, *snora27* or 300 pg human *DANCR* significantly increased the proportion of Class I embryos with severe phenotypic abnormalities (Fig 6D). While we do not yet know the mechanism behind this, these results suggest that *snora26* and *snora27* are biological relevant transcripts that can contribute to the developmental function of the *dancr* locus and that human *DANCR* can modify normal development in zebrafish.

## Discussion

LncRNA sequence and transcription is rapidly turned over during evolution limiting the use of comparative studies to identify biologically important transcripts based on primary sequence similarity alone [8]. As such, alternative approaches such as conservation of lncRNA genomic position relative to neighbouring protein coding genes and lncRNA subcellular localisation have been used to identify lncRNAs that act using similar mechanisms-of-action in distant species. To this end, syntenic lncRNAs possessing conserved functions in vertebrate development despite limited or no sequence similarity have been discovered and some of these have been implicated in cancer biology [14]. To search for new lncRNA regulators of melanoma biology and vertebrate development we have now identified 506 human candidate melanoma-associated lncRNAs whose loci are bound by MITF and that are transcribed from equivalent regions in the zebrafish genome. Our results demonstrate that one of these, *DANCR*, acts as an exemplar illustrating the potential importance of this set of lncRNAs. It displays robust positional synteny among diverse vertebrates, hosts two small RNAs within separate introns, is required for early embryonic development in zebrafish and promotes proliferation and migration in human melanoma.

MITF and c-MYC regulate *DANCR* in human melanoma. *DANCR* levels correlate with essential growth regulatory genes including c-MYC across CCLE cancer cell lines and it is highly expressed in many different types of cancer [34,35]. This indicates that *DANCR* may act within key proliferation inducing gene regulatory networks as a pan-cancer oncogene. Melanoma tumours are highly heterogeneous and comprise at least six different cell subpopulations with distinct biological

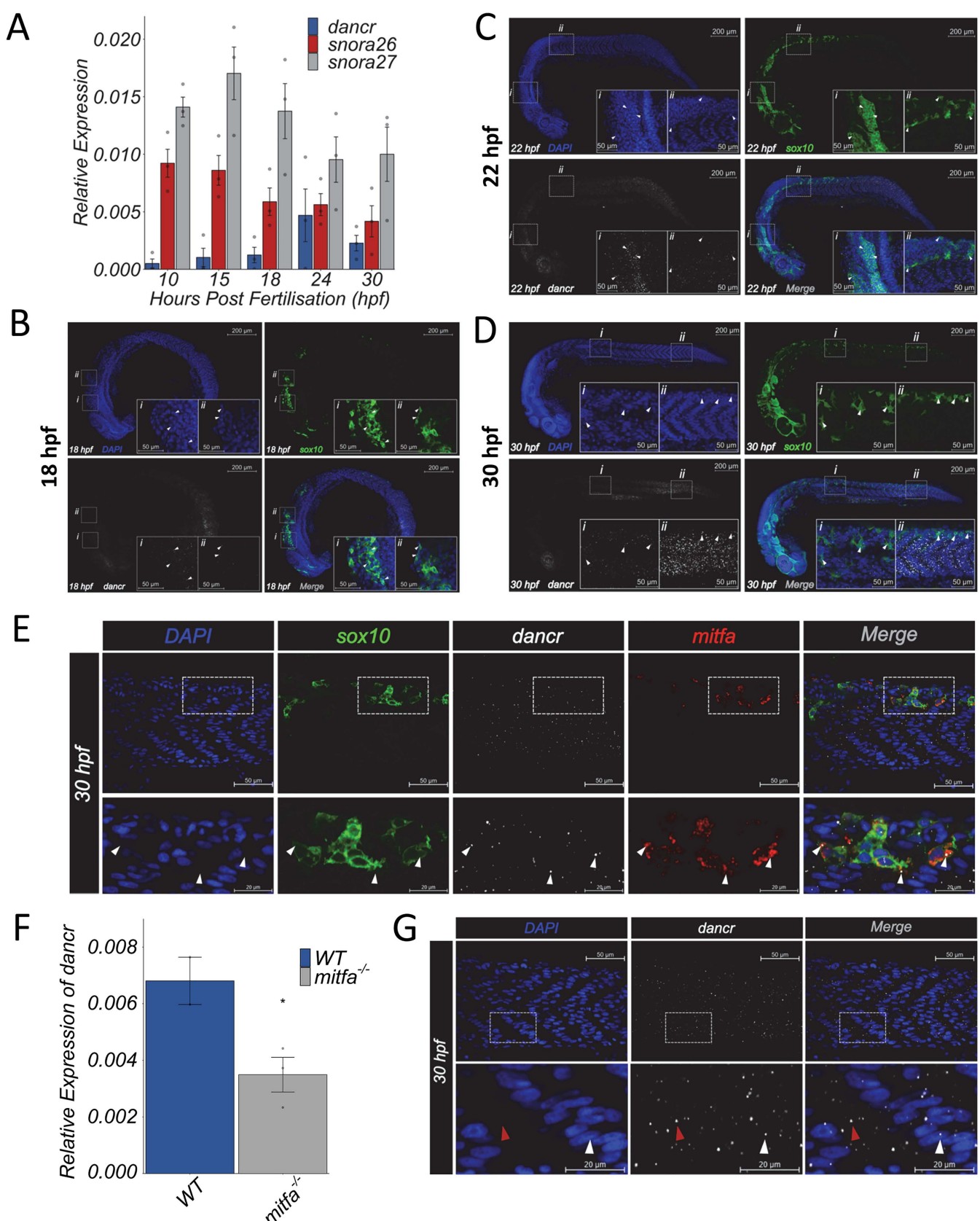

**Fig 4.  *Dancr* is dynamically expressed in multiple tissues and cell types in the developing zebrafish embryo.** *Dancr*, *snora26* and *snora27* are expressed during early embryonic development in zebrafish. **(A)** *Dancr, snora26* and *snora27* expression levels were measured in whole zebrafish embryos at the indicated developmental stages between 10-30 hpf using RT-qPCR. *actb2* was used as a reference gene. Spearman rank correlation showed that there is no association between expression of *dancr* and either *snora26* (R = 0.064, p = 0.82) or *snora27* (R = 0.054, p = 0.85). **(B, C, D)** *Dancr* is dynamically expressed in multiple tissues and cell types during zebrafish embryogenesis. Zebrafish *dancr* RNA transcripts were detected using RNAscope ISH on fixed embryos at 18 hpf **(B)**, 22 hpf **(C)** and 30 hpf **(D)**. Embryos are shown in lateral view. Confocal imaging of *dancr* (white), GFP expressing NCCs (green), nuclear DAPI (blue) and merged channels are shown. A 20X objective was used to obtain the representative images presented. Each representative image is derived from an individual Z-slice from the same embryo. White arrow heads indicate examples of neural crest cells expressing *dancr*. Scale bar = 200µm. Inset scale bar = 50µm. **(E)** A subset of melanoblasts co-express *mitfa* and *dancr*. RNAscope ISH detection of *dancr* (white) and *mitfa* (red) and immunofluorescence labelling of *sox10* (GFP) positive NCCs and derivatives (green) in the tail of 30hpf zebrafish embryos. A 63X objective was used to obtain representative images from an individual focal plane in a 3D projection. Embryos are presented laterally with the orientation of the head to the left. White arrow heads indicate examples of co-expression of *mitfa* and *dancr* within *sox10* positive melanoblasts. Scale bar = 50µm. Inset bar = 20µm. **(F)** *Dancr* expression was measured in *nacre* mutant (*mitfa*⁻/⁻) and wild type whole zebrafish embryos at 30 hpf using RT-qPCR. *actb1* was used as a reference gene. **(G)** *Dancr* transcript is enriched in the cytoplasm of cells in the developing zebrafish development. RNAscope ISH detecting *dancr* (white) and DAPI staining (blue) in the tail region of a single embryo at 30 hpf using a 63X objective. Embryos are shown in lateral view. Representative confocal image derived from an individual focal plane. The white and red arrowheads indicate nuclear and cytoplasmic *dancr* expression respectively. Scale bar = 50µm. Inset bar = 20µm.

phenotypes [42]. The ability of these cell populations to transition between different cellular states drives melanoma growth and metastasis and is a major barrier to the effectiveness of current treatments. Whilst MITF acts as a rheostat to govern the generation of different melanoma cell states in response to changes in the microenvironment, broad *DANCR* expression suggests additional modes of regulation. *DANCR* is highly expressed in all known cell states in melanoma [23] and in all four tumour genomic subtypes suggesting that it may play an essential role in melanoma irrespective of tumour heterogeneity and genomic status. Our finding that *DANCR* is directly regulated by both MITF and c-MYC may explain its high expression in both proliferative (MITF-high) and invasive (c-MYC-high) transcriptional states in melanoma. Therapeutic targeting of *DANCR* would therefore be predicted to block the growth of all melanoma cell states and may have important implications for the development of new treatments targeting drug-tolerant cell subpopulations to prevent tumour relapse.

Despite the recognised importance of *DANCR* in cancer, its function in normal development and physiology is less well understood. Our results now show that *dancr* is an essential regulator of early vertebrate development. *Dancr* promoter deletion in zebrafish led to multiple defects in the development of the anterior to posterior axis where the most severely affected mutant embryos lacked clear trunk and tail tissues and had reduced head sizes. This represents one of the most profound loss-of-function phenotypes for a lncRNA in zebrafish development and shows at least a superficial resemblance to the *snailhouse*/*bmp7a* mutant phenotype [43]. This indicates a possible contribution from disrupted BMP signalling during gastrulation, but the severity of the phenotype indicates that multiple embryological processes are likely disrupted; dissection of the mechanistic basis for the phenotype will be the subject of future studies.

*Dancr* is expressed in multiple locations in the developing zebrafish embryo, including the eyes, somites, forebrain, midbrain, hindbrain, NCCs and presumptive melanoblasts. *Dancr* expression is activated by *mitfa* during early zebrafish development indicating that both human and zebrafish *DANCR* are components of the MITF network. Its transcript levels decrease in cranial NCCs between 22 hpf and 30 hpf so that *dancr* expression becomes restricted to trunk NCC populations. As NCC development begins anteriorly and progresses along the anteroposterior axis [44], this progression in *dancr* expressing NCCs is consistent with a transient expression in premigratory and early migratory stages, and draws parallels to that found in human cell models of lineage differentiation where *DANCR* is highly expressed in somatic epidermal, adipocyte and osteoblast progenitor cell populations and downregulated upon lineage differentiation [31]. Moreover, *DANCR* knockdown in human organotypic epidermal tissue induces expression of differentiation genes [31]. This suggests that *dancr* may have a conserved developmental function to inhibit differentiation and maintain stem cell-like properties of progenitor cell populations. We speculate that *DANCR* may act in a similar way in cancer stem-cells, such as the drug

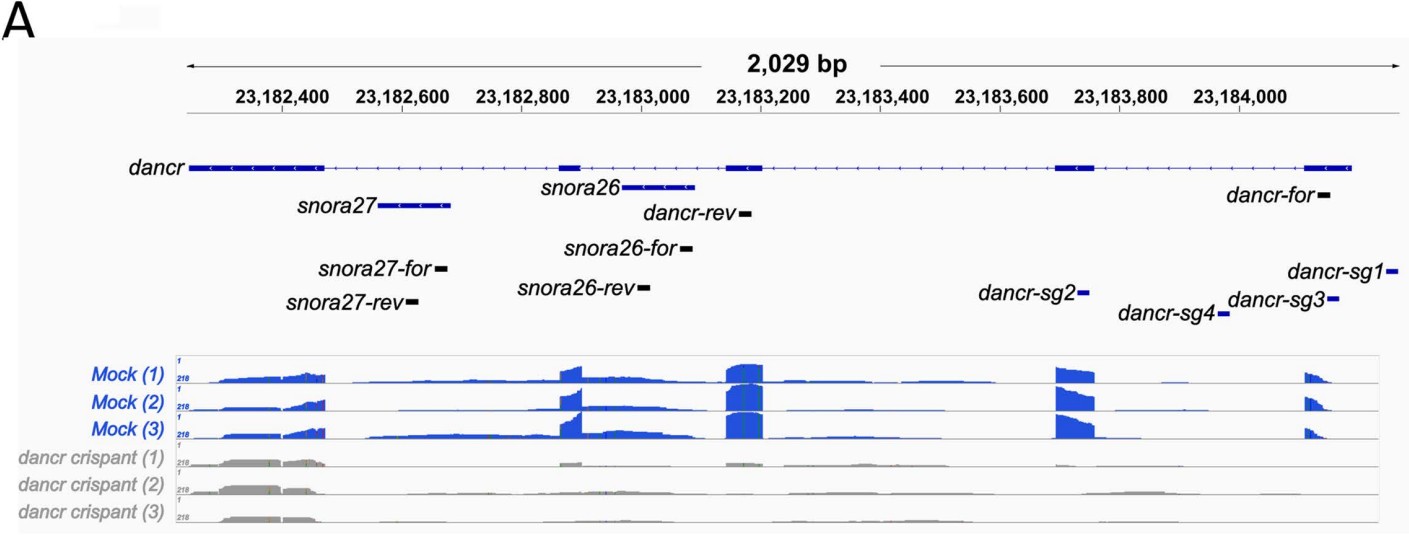

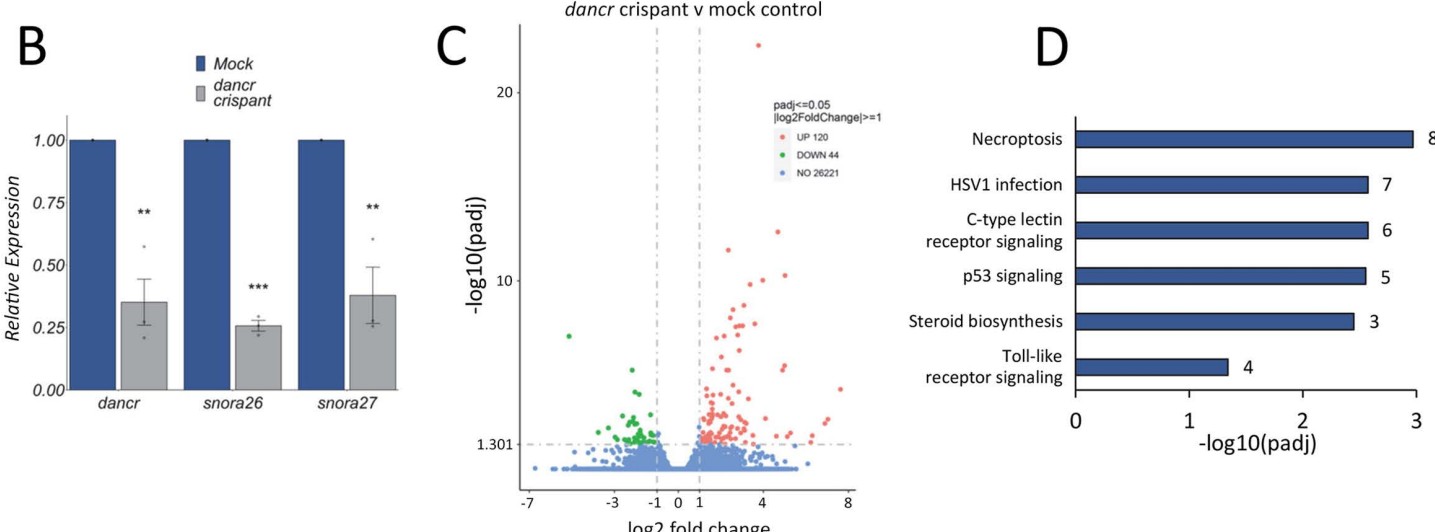

**Fig 5. *Dancr* regulates gene expression pathways involved in cell death.** CRISPR-Cas9 mediated deletion of the zebrafish *dancr* promoter leads to a reduction in *dancr*, *snora26* and *snora27* expression. **(A)** Integrative Genomics Viewer visualisation of RNA-seq reads mapping to the *dancr* locus in mock control and F0 dancr crispant embryos at 30 hpf. The locations of the four guide RNAs targeting the *dancr* promoter and primers used for RT-qPCR are shown. **(B)** *Dancr, snora26* and *snora27* expression levels were measured in F0 dancr crispant and mock control embryos at 30 hpf using RT-qPCR. *actb2* was used as a reference gene. Results presented as mean±sem, n=3. Two-tailed two sample t-test p<0.01**. Individual dots represent separate biological replicates. **(C, D)** *Dancr* regulates gene pathways involved in cell death and the immune response in the developing zebrafish embryo. **(C)** *Dancr* promoter deletion induces statistically significant changes in the expression of 164 genes (DESeq2 padj<=0.05; log2Fold-Change>=1.0). **(D)** KEGG pathway analysis of the *dancr* regulated gene set identified significantly enriched pathways (FDR 5%). The number of genes found in each category are indicated.

tolerant neural crest stem cell-like populations in melanoma, and that *DANCR* dependent re-activation of developmental programmes associated with NCC development could contribute to melanoma growth and metastasis.

LncRNA subcellular localization is critical for function and the same lncRNA can act differently in the nucleus and the cytoplasm through association with distinct sets of nucleic acid and protein targets [45–47]. This is significant as

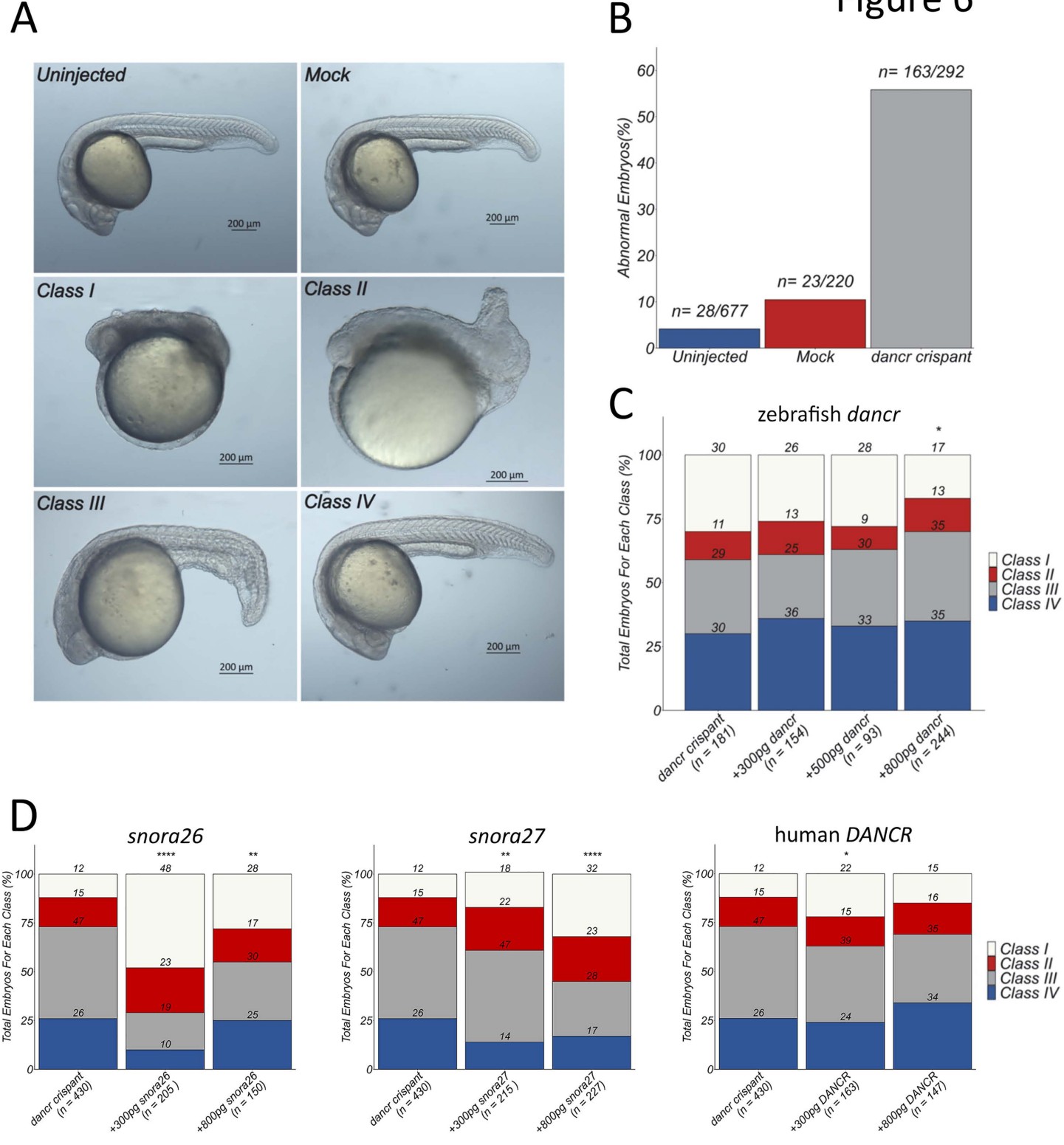

**Fig 6. _Dancr_ is essential for zebrafish embryogenesis. (A, B)** _Dancr_ del F0 crispant embryos display multiple overt developmental defects. The _dancr_ promoter was deleted in zebrafish using CRISPR-Cas9 genome editing and F0 crispant embryos were analysed at 24 hpf. **(A)** Phenotypic

abnormalities were separated into four classes based on their severity and representative images of the different classes are shown. Embryos are presented laterally. Scale bar = 200μm. **(B)** The number of abnormal embryos was quantified for *dancr* F0 crispant, uninjected wild type sibling and mock control embryos. Results are presented as a percentage of total embryos. **(C)** Zebrafish *dancr* transcript partially rescues the effect of *dancr* promoter deletion whilst **(D)** *snora26*, *snora27* and human *DANCR* co-injection increase the severity of the F0 *dancr* crispant phenotype. The indicated amounts of *in vitro* transcribed lncRNA or snoRNA were co-injected with the Cas9-crRNA-tracrRNA RNP complex in a genome editing experiment. The total number of F0 *dancr* crispant embryos in each class was quantified at 24 hpf and presented as a percentage of total embryos. Two-tailed two sample t-test $p < 0.05$*, $p < 0.01$**, $p < 0.001$***, $p < 0.0001$****.

many syntenic lncRNAs display different subcellular localisation patterns in human and mouse stem cells and do not have conserved functions [48]. *DANCR* is highly expressed compared to most lncRNAs and whilst both human and zebrafish *DANCR* transcripts are enriched in the cytoplasm they also display significant nuclear expression. Accordingly, *DANCR* has been reported to recruit chromatin regulators and modulate the activity of transcription factors in the nucleus [49,50], as well as act as a competitive endogenous RNA and regulator of mRNA stability and translation in the cytoplasm [51]. Our results do not discriminate between these scenarios and indicate that human and zebrafish *DANCR* may function as regulators of genes involved in shared biological processes and pathways crucial for development and dysregulated in cancer. Consistent with this, we found that both human and zebrafish *DANCR* regulate common pathways involved in cell death such as the p53 signalling pathway as well as some related genes. This includes the *CDKN1A* mediator of p53-dependent cell cycle arrest and apoptosis that was identified as a common target in both loss-of-function models, as well as caspase family members (*CASP7* in human; *casp8* in zebrafish) and matrix metalloproteinase genes involved in skin biology, apoptosis and melanoma metastasis (*MMP8* and *MMP15* in humans; *mmp9* and *mmp13a* in zebrafish).

Our discovery that the zebrafish *dancr* transcript partially rescued whilst human *DANCR* increased the severity of the zebrafish loss-of-function phenotype could be due to dosage sensitivity or suggest that human and zebrafish *DANCR* may have different mechanisms of action. Future interactome experiments will directly test this. Whilst these data also indicate that the developmental function of the locus is mediated at least in part by the fully spliced mature *dancr* lncRNA, *DANCR* is a member of a sub-class of lncRNAs that act as host genes for small non-coding RNAs and are predominantly localised in the cytoplasm and in general more widely expressed compared to most lncRNAs [52]. The small RNA genes within *DANCR* are located within two small patches of vertebrate DNA sequence conservation in separate introns. In zebrafish, both are snoRNAs, *snora26* and *snora27*, whilst in humans *snora27* appears to have evolved into the *MIR4449* miRNA. Accordingly, the processing of intronic snoRNAs and miRNAs is very similar and a subset of snoRNAs seem to have lost snoRNA functions and gained miRNA capabilities during evolution [53]. siRNA mediated depletion of *DANCR* targets the mature processed lncRNA transcript and did not affect the levels of *SNORA26* suggesting that the embedded small RNAs do not play a significant role in controlling proliferation and migration in melanoma. However, co-injection of *snora26* and *snora27* in F0 crispants increased the proportion of embryos with severe phenotypic abnormalities indicating that they are biologically important molecules that may contribute to the developmental function of the *dancr* locus. These are H/ACA box snoRNAs that are predicted to guide RNA pseudouridylation and their significance will be explored in future studies.

We have identified a set of syntenic lncRNAs exemplified by *DANCR*, a multi-exonic, cytoplasmically enriched lncRNA in both human and zebrafish. *DANCR* is a melanoma-associated lncRNA and conserved small non-coding RNA host gene that is expressed from equivalent regions in diverse vertebrate genomes and in a similar direction relative to the neighbouring *USP46* and *RASL11B* protein coding genes. Such syntenic lncRNAs may have conserved gene regulatory functions in vertebrate development and may be important in melanoma and other cancers.

## Methods

### Ethics Statement

This study was performed with approval by the Ethics Committee from the University of Bath and in accordance with the Animals Scientific Procedures Act (ASPA) 1986 under the Home Office Project Licence P87C67227.

### Identification of human-zebrafish syntenic lncRNAs

Zebrafish Ensembl 101 annotations together with RNA-seq data from early development [54], ageing time points [55], adult heart, brain, liver, muscle, blood [56] and melanocyte and melanoma cells [57,58] were used to assemble zebrafish transcript models. Pipeline for lncRNA annotation from RNA sequencing data (PLAR) was then performed as described in [6] to generate a set of putative zebrafish lncRNAs. These were compared to our previously identified MITF-bound human melanocyte and melanoma expressed lncRNA dataset [22] to identify syntenic lncRNAs expressed from equivalent regions in the human and zebrafish genomes.

### Transcriptomics

For gene expression analysis in human melanoma cells, total RNA was prepared in triplicate from *DANCR* knockdown and control SK-MEL-28 cells using the GeneJET RNA purification Kit (ThermoFisher Scientific). Residual genomic DNA was then removed using DNA-free DNA Removal Kit (ThermoFisher Scientific). PolyA selected 150-bp paired end (PE150) RNA sequencing was performed on the Illumina HiSeq4000 (Novogene). A minimum depth of 30M mapped reads were generated per sample. Differential gene expression and Gene Ontology analyses were performed as described in [22]. For zebrafish RNA-sequencing, approximately 30 mock control and Class I and II *dancr* del F0 crispant embryos in each group were homogenised using TRIzol (Invitrogen) and total RNA was isolated using the Quick-RNA Microprep Kit (Zymo Research). PolyA selected PE150 sequencing was performed on the Novoseq X Plus to a minimum depth of 50M reads per sample. Three biological replicates were used in each case. Novogene Bioinformatics pipelines were used to align processed reads to the GRCz10/danRer10 genome with HISAT2, call differentially expressed genes using DESeq2 and perform KEGG pathway analysis.

### Plasmid construction

sgRNAs targeting the *DANCR* promoter region were designed and cloned into pX-dCas9-mod-KRAB to generate plasmids for CRISPRi as described in [22]. For zebrafish *dancr* rescue experiments, full length *dancr* was synthesised as a gBlock and inserted as an XhoI fragment into pCS2+ (kindly provided by Dr Nikolas Nikolaou, University of Bath). The *dancr* gBlock sequence corresponds to the danRer11 RefSeq Gene LOC100536039 with a *5'-AGAC-3'* leader sequence and *5'-CTCGAG-3'* XhoI restriction site added to both ends. Oligonucleotides used in cloning are shown in S5 Table.

### Cell culture and transfections

Human 501mel, SK-MEL-28 and A375 melanoma cells were cultured in a humidified incubator in 5% $CO^2$ at 37°C in RPMI 1640 (Merck) supplemented with 10% foetal bovine serum (FBS; Gibco). For siRNA mediated knockdown (*MITF, MYC* and *DANCR*), approximately 1 x$10^5$ human melanoma cells were seeded in a 6-well plate. The next day, cells were transiently transfected using Lipofectamine 2000 (Invitrogen) as per the manufacturer's instructions using 100pmol of siRNA per well (siRNA sequences in S5 Table). Cells were harvested for downstream analysis 3 days after transfection. For CRISPRi mediated *DANCR* knockdown, roughly 1 x$10^5$ human melanoma cells were seeded in 6-well plate. The next day, 2µg plasmid DNA was transfected per well using Lipofectamine 2000 as per manufacturer's guidance. After three days, cells were trypsinised and resuspended in growth medium containing 0.7 µg/ml puromycin. Transfected cells were then seeded in a 6-cm dish, grown for 7 days in the presence of puromycin and harvested as a pool for downstream analysis.

## RT-qPCR

Total RNA was isolated from human melanoma cell lines using the GeneJET RNA purification Kit (ThermoFisher) and from zebrafish embryos using TRIzol (Invitrogen) extraction and Quick-RNA Microprep Kit (Zymo Research) purification as per the manufacturer's guidelines. RNA samples were reverse transcribed using the QuantiTect Reverse Transcription Kit (Qiagen) and qPCR was performed using Fast SYBR Green and the Step One Plus Real-Time PCR System (Applied Biosystems). Oligonucleotides used are shown in S5 Table.

## Cellular fractionation

Cellular fractionation of SK-MEL28 melanoma cells was performed as previously described in [22]. For biochemical cell fractionation of embryonic zebrafish cells, 30 embryos at 30 hpf were used. Embryos were manually dechorionated, embryo medium removed and collagenase/trypsin EDTA (20mg/ml) was added. Samples were incubated for 15 minutes at 30°C with shaking, then centrifuged at 4°C for 5 mins at 300g. Cells were re-suspended in 200μl of Lysis Buffer (10mM Tris HCl, pH 7.5, 150mM NaCl, 0.15% NP40, 0.5mM protease inhibitor cocktail (Roche) and 100U/ml RNAsin (Promega)) and incubated on ice for 20 mins. Cells were homogenised by passing through a 27G needle and placed on top of a 2.5 volume sucrose cushion. Samples were centrifuged at 4°C for 15 minutes at 20,000g. The cytoplasmic fraction (supernatant) and nuclear fraction (cell pellet) were separated, and RNA extracted for RT-qPCR.

## Proliferation and Migration Assay

For the proliferation assay, $1.5 \times 10^4$ siRNA or CRISPRi *DANCR* knockdown melanoma cells were seeded in a 6-well plate in growth medium (supplemented with 0.4 μg/ml of puromycin for the CRISPRi transfection). The total number of cells were counted at Day 0, 3 and 5 for the siRNA knockdown and Day 0, 2 and 4 for CRISPRi experiment using the Countess 3 FL Automated Cell Counter (Invitrogen).

For the migration assay, siRNA transfected SK-MEL-28 cells were mitotically inactivated using mitomycin C (Merck) to a final concentration of 4 μg ml$^{-1}$. Treated cells were incubated for 3 hours at 5% $CO_2$ and 37°C. Cell migration was investigated using 2 well culture inserts (Ibidi). These inserts comprise of two wells separated by a thin wall of 0.5mm to generate a gap. Inserts were placed in a 12-well plate and a total of 70μl of cells at a density of $4 \times 10^4$ were seeded into each side of the chamber. Cells were allowed to attach and were incubated until the next morning. Inserts were carefully removed with sterile tweezers and cells were washed twice with 1X PBS and growth medium was added. Images were taken using the EVOS FL microscope (Invitrogen) at 0hr, 24hr, and 48hr. Migration capacity was measured by calculating the area between the two migrating cell fronts using ImageJ [59] and the wound healing plugin (https://github.com/MontpellierRessourcesImagerie/imagej_macros_and_scripts/wiki/). The equation used to calculate percentage gap closure is described below where T0 represents the area calculated at 0 hours and Tx represents the area calculated at either 24 or 48 hours.

$$\% \; Gap \; Closure = \frac{(T0 - Tx)}{T0} \times 100$$

## Zebrafish maintenance

Adult zebrafish were housed within the University of Bath Fish Facility on a 14hr: 10hr light: dark cycle and embryos were obtained from natural crosses. Staging of embryos was performed following [60].

## Whole Mount Fluorescent in situ hybridisation and immunofluorescence

Wild type AB zebrafish were used throughout this work unless stated. *Tg(Sox10:Cre)ba74; Tg(hsp70l:loxP-dsRed-loxP –Lyn-Egfp)* zebrafish were used for the RNAscope experiments. Upon heat shock of this line, NCCs are labelled with

a membrane tethered GFP driven by the Cre-loxP system under the control of the *sox10* promoter [40,41]. Embryos were anesthetised and fixed in 4% paraformaldehyde (PFA; Alfa Aesar) at room temperature for 3 hours. PFA was then removed, 100% methanol was added and samples were stored at −20°C. The RNAscope Multiplex Fluorescent kit V2 (Bio-techne) was used as described in [41] and zebrafish *dancr* RNAscope probes were designed against RefSeq Gene ID LOC100536039. Briefly, methanol was removed and samples were air dried at room temperature for 30 mins. Proteinase Plus was added and incubated at room temperature for varying lengths of time depending upon developmental stage of the embryo. Samples were washed with 0.01% PBS-Tween and incubated overnight with the probes (*mitfa-C1* and *dancr-C3*). Probes were mixed following manufacturer guidelines. Samples were washed with 0.2X SSCT and addition of AMP1–3 was performed following manufacturer instructions. HRPC1 and HRPC3 was added with Opal 650 (1:2500; Akoya) and Opal 520 (1:2500; Akoya) respectively followed by HRP blocker. Washes were performed with 0.2X SSCT in between steps. For immunofluorescence detection of GFP positive NCCs, samples were incubated overnight at 4°C with rabbit anti-GFP primary antibody in blocking solution (1:750; Invitrogen). Blocking solution consisted of 5% goat serum (Vector Laboratories) and 1% DMSO (Sigma-Aldrich) in 0.1% PBS-Tween. Samples were rinsed with 0.1% PBS-Tween and incubated for 3 hours at room temperature with the goat anti-Rabbit Alexa Fluor488 (1:1000; Invitrogen) secondary antibody. Samples were washed with 0.1% PBS-Tween and DAPI (2mg/ml) diluted in blocking solution (1:1000; Roche) was added. Samples were stored in 0.1% PBS-Tween at 4°C and mounted in 50% glycerol/PBS. Images were acquired using the Zeiss LSM880 Confocal Microscope (Zeiss) with either a 20X objective or a 63X oil objective.

### Generation of F0 crispant zebrafish embryos

For generation of F0 crispants, a multiple guide CRISPR/Cas9 strategy was employed [61,62]. Briefly, four CRISPR RNAs (crRNAs) were designed using predicted on-target and off-target scores provided by CRISPR-Cas9 guide RNA (gRNA) design checker (Integrated Design Technologies) and CHOPCHOP algorithms [63]. Equimolar concentration of each individual crRNA and tracrRNA were combined, diluted to 61μM using Nuclease Duplex Buffer (Integrated DNA Technologies) and heated to 95°C for 5 mins before being cooled on ice for 2 mins. Mock injection mixtures lacked the *dancr* targeting crRNA. Individual gRNAs and Alt-R S.p. Cas9 Nuclease V3 (Integrated DNA Technologies) were combined to a 1:1 ratio and heated to 37°C for 5 mins to form ribonucleoprotein (RNP) complexes. Four RNPs per lncRNA loci were pooled together in equal volumes to a final concentration of 30.5μM. 2nl mixture was injected into the yolk of embryos at the single cell stage.

### *In Vitro* synthesis of capped RNA

Full length capped RNA for rescue experiments were produced using the mMESSAGE mMACHINE SP6 Transcription Kit (Ambion) as per the manufacturer's instructions. Plasmids were linearised using NotI (ThermoFisher Scientific) and purified using the Monarch PCR & DNA Cleanup Kit (New England Biolabs) as per manufacturer's instructions. For the SP6 transcription reaction, 1μg of linearised plasmid was added as per manufacturer's instructions. TURBO DNase (Ambion) was added to remove any remaining DNA. Synthesised RNA was purified using the MEGAClear Kit (Ambion).

### Statistics

Statistical analysis and production of graphs was performed using R statistical software, version 4.2.3 (https://www.r-project.org/). Data is presented as the mean ± standard error. The normality of the data was assessed by the Shapiro-Wilks test. For statistical analysis, either a two-tailed two sample t-test was performed or a one-way ANOVA followed by post-hoc Tukey's test. The type of statistical analysis performed is indicated, where applicable, in each Fig. For all analyses, $p < 0.05$ was deemed as significant.

## Supporting information

**S1 Fig. Validation of *DANCR* target genes and effects of siRNA mediated depletion of *DANCR* on *SNORA26*.** (A) Principal component analysis (PCA) shows that control and si*DANCR* knockdown samples cluster separately. (B, C) *DANCR* was depleted in A375 cells using two independent siRNAs. Three days later *DANCR* and (B) the expression of the indicated *DANCR* target genes or (C) *SNORA26* levels were determined using RT-qPCR. *POLII* was used as a reference gene. Results are presented as mean+/- SEM., n=3. Two-tailed two sample t-test p<0.05. Individual dots represent separate biological replicates. (TIF)

**S2 Fig. *DANCR* promotes A375 human melanoma cell proliferation and migration.** *DANCR* expression was silenced in A375 cells by dCas9-KRAB mediated CRISPRi using two independent sgRNAs targeting the *DANCR* promoter. Three days later *DANCR* levels were measured using RT-qPCR and proliferation (A, B) or wound healing (C, D, E) assays set up. Expression changes are shown relative to a non-targeting control sgRNA (set at 1). *POLII* was used as a reference gene. For proliferation analysis, cells were seeded in a 6-well plate and the total number of cells were counted at days 0, 2 and 4 (B). For wound healing assays, cells were first treated with mitomycin-C to block cell proliferation and migration was then determined using Ibidi chambers (Culture-Inserts 2 Well). The gap was imaged at 0, 24 and 48 hours and percentage gap closure calculated using the ImageJ Wound Healing plugin (D, E). Statistical analysis was performed at the 48-hour time point. All results presented as mean+/- SEM., n≥2. Two-tailed two sample t-test p<0.05*, p<0.01**, p<0.001***, p<0.0001****. Individual dots represent separate biological replicates. (TIF)

**S3 Fig. Sequence conservation analysis and validation of *dancr* promoter deletion in F0 *dancr* crispants.** (A) Genome browser view of the human *DANCR* locus with the 5' end of a representative isoform and alignment of 100 vertebrate species. Splice sites with positive PhyloP scores are shaded and a predicted MYC binding site from the JASPAR database is shown. Bottom: Genome browser view of zebrafish *dancr* as annotated in RefSeq displaying the snoRNA positions from Ensembl, MITF binding site motif and alignment of 8 fish species. (B) Genome browser view displaying the zebrafish *dancr* locus (GRCz10/danRer10). The location of the sgRNAs used to guide CRISPR/Cas9 mediated deletion of the *dancr* promoter and PCR primers flanking the targeted deletion site for screening are shown. Alignment of representative PCR product sequences indicate the position of the *dancr* promoter deletions. (C) PCR amplification of the genomic region flanking the proposed *dancr* promoter deletion was performed using genomic DNA extracted from individual *dancr* crispant, uninjected and mock control embryos at 24 hpf. A negative control containing water instead of genomic DNA was also used. PCR products were analysed by agarose gel electrophoresis. An expected band of 725 bp corresponding to wild type sequence was amplified in the uninjected and mock control embryos. Multiple bands of varying sizes, due to the mosaic deletions caused by multiple sgRNAs targeting the *dancr* promoter, were generated in the F0 *dancr* crispant embryos. (D) PCA analysis on the gene expression value (FPKM) separates the control and knockdown samples. (TIF)

**S1 Table. 2,796 syntenic human lncRNAs that have a positionally equivalent transcript in zebrafish.** (XLSX)

**S2 Table. 506 syntenic lncRNAs that contain a MITF ChIP-seq binding site in humans.** (XLSX)

**S3 Table. *DANCR* regulated genes in SK-MEL-28 cells.** (XLSX)

**S4 Table. *Dancr* promoter deletion in zebrafish leads to significant changes in 164 genes.** (XLSX)

**S5 Table. Sequence of oligonucleotides used in this study.**
(XLSX)

**S6 Table. Numerical data underlying the graphs.**
(XLSX)

## Acknowledgments

We thank University of Bath Final Year Undergraduate Project students (Milly Cooke, Caitlin Davies) for help with the proliferation assays and Prof Adele Murrell for critically reading the manuscript.

## Author contributions

**Conceptualization:** Stephanie M.E. Jones, Robert N Kelsh, Keith W Vance.

**Data curation:** Stephanie M.E. Jones, Keith W Vance.

**Formal analysis:** Stephanie M.E. Jones, Michael Shapiro, Igor Ulitsky, Robert N Kelsh, Keith W Vance.

**Funding acquisition:** Nikolas Nikolaou, Keith W Vance.

**Investigation:** Stephanie M.E. Jones, Elizabeth A Coe, Michael Shapiro, Kelli M Gallacher, Karen Camargo Sosa, Igor Ulitsky, Keith W Vance.

**Methodology:** Stephanie M.E. Jones, Igor Ulitsky, Robert N Kelsh, Keith W Vance.

**Project administration:** Keith W Vance.

**Resources:** Stephanie M.E. Jones.

**Software:** Igor Ulitsky.

**Supervision:** Nikolas Nikolaou, Robert N Kelsh, Keith W Vance.

**Writing – original draft:** Keith W Vance.

**Writing – review & editing:** Stephanie M.E. Jones, Elizabeth A Coe, Nikolas Nikolaou, Robert N Kelsh, Keith W Vance.

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
