## [Decision Letter · Decision Letter 0]

16 Jul 2025

PGENETICS-D-25-00452

The syntenic long non-coding RNA DANCR is an essential regulator of zebrafish development and a human melanoma oncogene

PLOS Genetics

Dear Dr. Vance,

Thank you for submitting your manuscript to PLOS Genetics. After careful consideration, we feel that it has merit but does not fully meet PLOS Genetics's publication criteria as it currently stands. Therefore, we invite you to submit a revised version of the manuscript that addresses the points raised during the review process.

Please submit your revised manuscript within 60 days Sep 14 2025 11:59PM. If you will need more time than this to complete your revisions, please reply to this message or contact the journal office at plosgenetics@plos.org. Please include the following items when submitting your revised manuscript:

We look forward to receiving your revised manuscript.

Kind regards,

Richard Mark White, MD, PhD

Guest Editor

PLOS Genetics

Monica Colaiácovo

Section Editor

PLOS Genetics

Aimée Dudley

Editor-in-Chief

PLOS Genetics

Anne Goriely

Editor-in-Chief

PLOS Genetics

**Journal Requirements:**

1) Please provide an Author Summary. This should appear in your manuscript between the Abstract (if applicable) and the Introduction, and should be 150-200 words long. The aim should be to make your findings accessible to a wide audience that includes both scientists and non-scientists. Sample summaries can be found on our website under Submission Guidelines:

https://journals.plos.org/plosgenetics/s/submission-guidelines#loc-parts-of-a-submission

- TM on pages: 21, and 23.

Potential Copyright Issues:

i) Please confirm (a) that you are the photographer of 6A, or (b) provide written permission from the photographer to publish the photo(s) under our CC BY 4.0 license.

5) Thank you for stating that "Zebrafish dancr crispant RNA-Seq data: Gene Expression Omnibus GSE292918(https://www.ncbi.nlm.nih.gov/geo/query/acc.cgi?acc=GSE292918)." We noted  that the dataset is currently private and is scheduled to be released on Mar 24, 2026. Please note that, though access restrictions are acceptable now, your entire minimal dataset will need to be made freely accessible if your manuscript is accepted for publication. This policy applies to all data except where public deposition would breach compliance with the protocol approved by your research ethics board. If you are unable to adhere to our open data policy, please kindly revise your statement to explain your reasoning and we will seek the editor's input on an exemption.

2) If any authors received a salary from any of your funders, please state which authors and which funders.

7) Please ensure that the funders and grant numbers match between the Financial Disclosure field and the Funding Information tab in your submission form. Note that the funders must be provided in the same order in both places as well. Currently, the order of the grants is different in both places. 

**Reviewers' comments:**

Reviewer's Responses to Questions

Reviewer #1: The paper entitled “The syntenic long non-coding RNA DANCR is an essential regulator of zebrafish development and a human melanoma oncogene” adds an important set of data to the misconception that lincRNAs are solely fine-tuners or not important at all for the organismal homeostasis. However, for publication a few points should be clarified. Find them below.

Major points:

-The in cellulo data the KO in the fish point towards a role of DANCR at least in part in the detected phenotype. It would be nice to describe which specific phenotypes are affected by the snoRNAs Vs DANCR. Are the phenotypes affected by DANCR related to melanocyte biology?

-In Figure 2B, the KD of MITF in SK-MEL-28 is unsuccessful; this is unlikely to be due to the specific cell line, as KD of MITF has been performed successfully by several groups in these line. The experiment should be repeated, possibly with validated siRNAs. Also, it is not clear why these cell lines were chosen, and an explanation for the different outcome of MITF KD in A375 and Mel501 is not provided. In this respect, it would be helpful to compare the relative expression levels of MITF and MYC in the different cell lines, whether CNA or mutations of DANCR exist, and whether they include or not the promoter region.

-Figure 3A: It is not clear whether this is one or more experiments and whether the KD is statistically significant.

-Figure 3D: From this picture, it looks like siRNA 2 is the most efficient at knocking down DANCR, is thus not clear the rational for using siRNA 1 for the sequencing.

-Figure 3E-H: To estimate the effect on proliferation and wound healing, also day 1 and 2 should be shown in the proliferation assay. This would also allow proper normalisation of the data in G and H. Knowing that invasion and proliferation in melanoma are mutually exclusive, I would not be surprised to discover that the effect on the closure of the wound is a consequence of the lack of cell proliferation.

-Mechanistically, the study concludes that DANCR regulates the expression of a good number of transcripts both in human melanoma lines and zebrafish, converging towards regulation of cell death and p53 signalling. Could the authors elaborate on the mechanism? Is this a function of the nuclear portion of the transcript? In that case, is the stoichiometry consistent with the hypothesis?

Minor:

-I detected many typos (e.g. page 9 CRIPSRi) and, especially in the discussion, a lot of references are missing (e.g. page 18 "DANCR knockdown in human organotypic epidermal tissue induced expression of differentiation genes." )

-It is not clear why a zebrafish developmental model was chosen over a zebrafish melanoma model.

Reviewer #2: The manuscript by Jones et al., describes the identification of a long non-coding RNA that the authors called DANCR. The DANCR lncRNA is conserved throughout vertebrate evolution at the genomic position, in respect to its protein-coding neighbouring genes, USP46 and RASL11b. DANCR is under the regulation of MITF and c-MYC transcription factors with the established role in melanomagenesis. Through a series of knock-down assays in human melanoma cell lines, the authors propose that DANCR acts in the cytoplasm to control cell proliferation and migration through the regulation of a set of cancer-associated genes. Furthermore, the authors explore the function of the Dancr syntologous transcript in development in distantly related zebrafish. The authors describe a profound effect of Dancr loss of function on early embryonic development suggesting that this lncRNA has an important biological function already in species at the base of vertebrate evolution.

The study has the potential of being of interest to the broad scientific audience. However, several important points below should be addressed.

Major concerns:

1. Phenotypic characterisation of Dancr loss of function in the zebrafish embryos:

Is there a specific reason for restricting analyses of the phenotypic consequences of Dancr loss to CRISPants (zebrafish embryos transiently expressing four guide RNAs targeting the promoter region of Dancr)? Whereas CRISPant analyses gain promising results, they are also prone to artefacts. Simultaneous usage of four different gRNAs also increases the risk of off targeting. A gold standard in the field is generation of stable mutants that should be backcrossed several times to outcross any potential off target mutations. The range of severity of phenotypes achieved in CRISPants strongly suggest that there are some artefactual effects, which should be clarified by generating stable mutants.

Whereas CRISPants offer an advantage of carrying out rescue experiments. However, the observed phenotypes should be confirmed in stable mutants. If confirmed, rescue experiments can be done using CRISPants.

2. The connection between human melanoma and zebrafish embryonic development: Without making a clear link between the two phenomenon, melanomagenesis and early embryonic development, the logic of the manuscript is not clear. The authors state that the human and zebrafish lncRNAs potentially regulate same genes. However, there is no direct evidence for it. One way to connect zebrafish development and human melanoma is to perform rescue experiments with the human DANCR transcript. It appears that the developmental effects observed in zebrafish are much more pronounced that the human effect of DANCR. This again suggests that the phenotype of transient CRISPants should be confirmed/clarified in stable mutants.

Additional specific comments:

3. Deletion of Dancr promoter region in zebrafish embryos is an important point that should be more elaborated in the manuscript. What is the expected size of the promoter region deletion? Figure 5A should show position of primers used in Figure 5B for qRT-PCR.

4. A clearer schematic presentation of genome editing strategy and genotyping would be very useful. The authors should provide also rationale for the usage of four gRNAs and not two gRNAs which are more standard.

5. In Figure 6A, the authors show a range of phenotypes achieved when embryos are transiently express four gRNAs targeting Dancr promoter. For the RNA-seq analysis on CRISPants (Figure 5C), embryos with which grade of phenotypic abnormalities were used? How many biological replicates were used for RNA-seq. The authors should show also PCA analyses.

6. Figure 3B: were GO term analyses done on both, significantly up-regulated and down-regulated genes? This should be indicated.

7. “We next investigated the function of the zebrafish orthologue of DANCR in development”: Those are not exactly orthologs but syntologs or positional orthologs. The authors should be clear about their nomenclature.

A detailed sequence comparison should be carried out to see if promoter region or some of the splice junctions are conserved between distantly related DANCR transcripts.

8. The authors should do a sub-cellular fractionation experiment also in zebrafish, not only in cell lines to conclude on Dancr subcellular localization in zebrafish. The single molecule in situ hybridisation has rather low resolution to fully support cytoplasmic localization of Dancr.

9. As the study does not provide any molecular insides into the mode of action of DANCR, sentences such as “…human and zebrafish DANCR may exert similar functions or work using similar mechanisms of action” should be removed or re-phrased.

10. Is zebrafish Dancr also under control of MITF and c-MYC?

Reviewer #3: This paper from Jones and collaborators aims to clarify the role of the DANCR lncRNA in melanoma and during zebrafish embryogenesis. The pro-tumorigenic role of DANCR has been demonstrated in several cancer types and some of the regulated pathways have been investigated by other works. However, the role of DANCR in melanoma has been poorly investigated and no data are available on zebrafish development. Therefore, the presented data are novel and interesting, although mechanistic characterization of the DANCR role at molecular level is superficial. The experiments are generally well conducted with relevant controls in place and the paper is well-written, with a clear rationale.

However, I have some specific issues that may be addressed before publication:

1. The Authors showed that DANCR is mainly localized in the cytoplasm, even if a small fraction is retained into the nucleus. Then, they used RNA-sequencing to characterize the consequences of DANCR knockdown in both melanoma cell line and zebrafish model. In my opinion, this approach would be more suitable for lncRNA having nuclear localization and putative transcription-regulating function. Indeed, they found only a small number of transcripts deregulated. The possible function(s) of DANCR in the nucleus may be better discussed to justify this choice and to further highlight the complexity of this lncRNA activities.

2. The extent of overlap between genes deregulated upon DANCR knockdown in melanoma cell line and in zebrafish may be clearly represented in Figure 5. They mentioned these genes in the discussion section, but they should be depicted somehow also in the Figure.

3. Validation by qRT-PCR of at least some of the deregulated genes is required (Figure 3 and Figure 5). A validation of genes deregulated in SK-MEL-28 also in a second melanoma cell line would add robustness to the entire work.

4. In Figure 6, the Authors show that CRISPR-mediated deletion of the DANCR promoter region induces defects in zebrafish embryogenesis. These defects are partially rescued by re-expressing the DANCR mature transcript. However, the Authors could not rule out the possible contribution of snora26 or snora27 elimination to the phenotype. Is it possible that deleting the DANCR promoter, the chromatin status of the entire genomic locus has been altered? In other words, is the expression of the neighbor genes, such as usp46 or rasl11b, altered upon DANCR promoter deletion?

**Have all data underlying the figures and results presented in the manuscript been provided?**

Reviewer #1: Yes

Reviewer #2: Yes

Reviewer #3: **No: ** Numerical data underlying the graphs in the Figures have not been provided.

PLOS authors have the option to publish the peer review history of their article (what does this mean? ). If published, this will include your full peer review and any attached files.

**Do you want your identity to be public for this peer review?** For information about this choice, including consent withdrawal, please see our Privacy Policy .

Reviewer #1: No

Reviewer #2: No

Reviewer #3: No

**Figure resubmission:**
---

## [Decision Letter · Decision Letter 1]

24 Nov 2025

Dear Dr Vance,

We are pleased to inform you that your manuscript entitled "The syntenic long non-coding RNA DANCR is an essential regulator of zebrafish development and a human melanoma oncogene" has been editorially accepted for publication in PLOS Genetics. Congratulations!

Yours sincerely,

Monica Colaiácovo

Section Editor

PLOS Genetics

Aimée Dudley

Editor-in-Chief

PLOS Genetics

Anne Goriely

Editor-in-Chief

PLOS Genetics

BlueSky: @plos.bsky.social

Comments from the reviewers (if applicable):

Reviewer's Responses to Questions

**Comments to the Authors:**

Reviewer #1: The story looks still not very coherent, meaning that the jump from development to melanoma is big, however I do not have any further comment.

Reviewer #2: The authors have fully addressed all of my previous concerns. The addition of new experimental data (including validation of CRISPant phenotypes in stable mutant lines, the human DANCR co-injection experiment, RNA-seq PCA analyses, promoter deletion mapping, and qPCR validation in a second melanoma cell line) substantially strengthens the manuscript. The revised text is clearer and more rigorous, and the overall study now presents a coherent and compelling case for the developmental and oncogenic significance of DANCR.

I have no remaining concerns and support publication.

I have only a few minor, non-essential suggestions the authors may consider for clarity:

1. Human DANCR co-injection

Since human DANCR increases phenotype severity in zebrafish, a short explanatory sentence discussing possible interpretations (e.g., dosage sensitivity, species-specific context) may help readers.

2. snora26/snora27 contribution

The worsening phenotype upon snoRNA co-injection is intriguing; a brief comment in the Discussion regarding potential biological mechanisms could strengthen the narrative.

3. Mechanistic framing

Adding a single sentence explicitly noting that precise nuclear vs cytoplasmic mechanisms will require future interactome studies would help frame the limits of the current mechanism.

These are optional refinements and do not affect my overall recommendation for acceptance.

Reviewer #3: The Authors perfomed new experimental work to answer reviewers questions. I have no further major concerns. However, I noticed in the newly added data in Figure 6 that the phenotype rescue experiment with human DANCR resulted in increased Class I developmental defetcs. This seems paradoxical and may indicate different mechanisms of action between human and zebrafish DANCR. Few lines may be added to Discussion to address this issue.

**Have all data underlying the figures and results presented in the manuscript been provided?**

Reviewer #1: Yes

Reviewer #2: Yes

Reviewer #3: Yes

PLOS authors have the option to publish the peer review history of their article (what does this mean? ). If published, this will include your full peer review and any attached files.

**Do you want your identity to be public for this peer review?** For information about this choice, including consent withdrawal, please see our Privacy Policy .

Reviewer #1: No

Reviewer #2: No

Reviewer #3: No

**Data Deposition**

http://datadryad.org/submit?journalID=pgenetics&manu=PGENETICS-D-25-00452R1

**Press Queries**

---

## [Editor Report · Acceptance letter]

29 Nov 2025

PGENETICS-D-25-00452R1

The syntenic long non-coding RNA DANCR is an essential regulator of zebrafish development and a human melanoma oncogene

Dear Dr Vance,

We are pleased to inform you that your manuscript entitled "The syntenic long non-coding RNA DANCR is an essential regulator of zebrafish development and a human melanoma oncogene" has been formally accepted for publication in PLOS Genetics! Your manuscript is now with our production department and you will be notified of the publication date in due course.

With kind regards,

Zsofia Freund

PLOS Genetics

On behalf of:
